# The Hippo–Yki Signaling Pathway Positively Regulates Immune Response against *Vibrio* Infection in Shrimp

**DOI:** 10.3390/ijms231911897

**Published:** 2022-10-07

**Authors:** Linwei Yang, Zi-Ang Wang, Ran Geng, Shengwen Niu, Hongliang Zuo, Zhixun Guo, Shaoping Weng, Jianguo He, Xiaopeng Xu

**Affiliations:** 1State Key Laboratory of Biocontrol, School of Life Sciences, Sun Yat-sen University, Guangzhou 510275, China; 2Southern Marine Science and Engineering Guangdong Laboratory (Zhuhai), Zhuhai 519000, China; 3Institute of Aquatic Economic Animals and Guangdong Province Key Laboratory for Aquatic Economic Animals, Sun Yat-sen University, Guangzhou 510275, China; 4South China Sea Fisheries Research Institute (CAFS), Guangzhou 510300, China

**Keywords:** *Penaeus vannamei*, Yki, Hippo, Wts, *Vibrio parahaemolyticus*, signaling pathway

## Abstract

In the Hippo pathway, activation of Hippo and Warts (Wts) kinases results in the phosphorylation of Yorkie (Yki), to prevent its nuclear translocation. Shrimp aquaculture is threatened by *Vibrio* genus bacteria. In this study, we examine the role of the Hippo pathway in immune defense against *Vibrio parahaemolyticus* in Pacific white shrimp *Penaeus vannamei*. We show that *V. parahaemolyticus* infection promotes the expression of Yki and facilitates the dephosphorylation and nuclear translocation of Yki, indicating the inhibition of Hippo signaling upon bacterial infection. There is a complex regulatory relationship between the Hippo pathway components Hippo, Wts, and Yki and the immune-related transcription factors Dorsal, Relish, and STAT. Silencing of Hippo and Wts weakened hemocyte phagocytosis, while the silencing of Yki enhanced it, suggesting a positive regulation of shrimp cellular immunity by Hippo signaling activation. In vivo silencing of Hippo and Wts decreased the survival rates of *V. parahaemolyticus*-infected shrimp and elevated the bacterial content in tissues, while the silencing of Yki showed the opposite results. This suggests that the activation of Hippo signaling and the inhibition of Yki enhance antibacterial immunity in shrimp.

## 1. Introduction

As an evolutionarily conserved intracellular signaling cascade in animals, the Hippo–Yki pathway regulates cell survival, proliferation, mobility, and differentiation, playing pivotal roles in the development and regeneration of organs [1,2,3]. The fundamental structure of the canonical Hippo–Yki signaling, first uncovered in *Drosophila*, consists of the cytoplasmic kinases Hippo and Warts (Wts) and a transcriptional co-activator Yorkie (Yki), which correspond to Mst1/2 (also called STK4/STK3), Lats1/2, and Yes-associated protein (YAP) in mammals, respectively [4,5,6]. In *Drosophila*, upon the inactivation of upstream Hippo signaling, the unphosphorylated Yki is translocated into the nucleus and functions to promote cell proliferation and resist apoptosis by aiding the transcription factor Scalloped (sd) and regulating the transcription of a series of downstream genes [5,7]. The activation of Hippo signaling is characterized by the cascade phosphorylation of Hippo and Wts, which phosphorylates Yki to retain it in the cytoplasm, leading to cell proliferation arrest or even death [8,9]. Similar regulation and signal transduction steps also occur in the mammalian Hippo–Yki signaling pathway [10]. In mammals, Hippo signaling is known to be involved in the immune response by regulating cytokine production and immune cell behaviors [11,12,13,14]. Abnormal activation or inactivation of Hippo signaling is correlated with the pathological processes of a diversity of diseases, in particular cancer and inflammation [15,16,17,18,19,20]. However, little is known about the roles of the Hippo pathway in invertebrates, especially in the immune system.

Pacific white shrimp, *Penaeus vannamei* (*Litopenaeus vannamei*), is the most farmed shrimp species worldwide [21]. At present, the shrimp farming industry is heavily threatened by a variety of pathogens [22,23], among which the Gram-negative bacterium *Vibrio parahaemolyticus* is one of the most important and common pathogenic bacteria of shrimp. *V. parahaemolyticus* is known to be the major infectious agent of a diversity of shrimp diseases, with pathological signs of black gill, acute hepatopancreas necrosis, and red leg [24,25,26]. Understanding the regulation mechanisms of shrimp antibacterial immunity is the basis for containing bacterial diseases. The current study initially explores the activation profile of Hippo signaling during bacterial infection and further investigates its role in the regulation of the antibacterial response against *V. parahaemolyticus*, providing a basis for the in-depth revelation of the regulatory mechanism of shrimp immune defense.

## 2. Results

### 2.1. Expression Profiles of Hippo Signaling Components

Expression of three key components of the Hippo–Yki pathway, Hippo, Wts, and Yki, in immune-stimulated shrimp was detected by qPCR. Compared with the control, after injection with *V. parahaemolyticus*, the Gram-positive *Staphylococcus aureus*, the fungus *Aspergillus niger*, and lipopolysaccharide (LPS), the expression trends of Hippo and Wts were similar, with a change range of less than 1.5-fold at most time points (Figure 1A,B). In contrast, the change in Yki expression was also limited after stimulations by *S. aureus*, *A. niger*, and LPS but significantly increased after *V. parahaemolyticus* infection, especially at 48 and 96 h post stimulation, at which its expression was 18.4 and 6.9 times higher than that of the control, respectively (Figure 1C). This indicates that Yki could be involved in the immune defense against *Vibrio* infection.

### 2.2. Activation of Yki upon Bacterial Infection

The activation of Hippo signaling leads to the phosphorylation and cytoplasmic retention of Yki. Western blot analyses demonstrated that after stimulations with *V. parahaemolyticus* and the Gram-positive bacterium *Bacillus thuringiensis*, the levels of phosphorylated Yki were significantly reduced compared with the control (Figure 2A). Concordantly, the level of Yki protein translocated from the cytoplasm to the nucleus was also significantly elevated after infection by these two bacteria (Figure 2B). Immunofluorescence assay further confirmed that the protein level of Yki in hemocyte nuclei was obviously increased after *V. parahaemolyticus* infection (Figure 2C). This indicates that the bacterial infection in shrimp leads to the suppression of Hippo signaling and the activation of Yki.

### 2.3. Regulatory Relationship between Hippo Signaling and Immune Signaling Pathways

The JAK-STAT and NF-κB pathways play central roles in shrimp antibacterial defense. In the promoters of Hippo, Wts, and Yki, multiple NF-κB- and STAT-binding sites were predicted, indicating the regulatory relationships between Hippo signaling and these transcription factors (Appendix A). The regulatory mechanisms of the transcription of these Hippo pathway components were explored using dual-luciferase gene reporter assays. The results demonstrated that the two NF-κB family members of shrimp had opposite effects on the Hippo promoter, the activity of which could be enhanced by Relish but reduced by Dorsal (Figure 3A). The mRNA level of Hippo was up-regulated after the silencing of Dorsal but did not change significantly after the silencing of Relish (Figure 3D). In contrast, STAT did not affect the activity of the Hippo promoter, but the silencing of STAT significantly reduced the mRNA level of Hippo in vivo. Similarly, the activity of the Wts promoter was also suppressed by Dorsal, and the mRNA level of Wts was down-regulated upon the silencing of Relish and STAT (Figure 3B,E). By contrast, the activity of Yki promoter could be negatively regulated by Dorsal but enhanced by Relish and STAT (Figure 3C). Consistently, the expression of Yki was up-regulated upon Dorsal silencing but down-regulated upon Relish and STAT silencing (Figure 3F).

The regulatory roles of Hippo signaling on NF-κB and STAT were further detected. In Hippo-silenced shrimp, the expression of Dorsal and Relish was down-regulated, but that of STAT was up-regulated (Figure 3G). In Wts-silenced shrimp, the expression of Dorsal was also down-regulated, but that of Relish and STAT did not change significantly (Figure 3H). Contrary to those in Hippo-silenced shrimp, the expression of Dorsal and STAT in Yki-silenced shrimp was significantly up-regulated and down-regulated, respectively (Figure 3I). However, the mRNA level of Relish was also decreased after Yki silencing, the trend of which is the same as that after Hippo silencing.

### 2.4. The Regulation of Hemocyte Phagocytosis by Hippo Signaling

Crustacean hemocytes are essentially implicated in the antibacterial response by directly phagocytizing foreign materials. The regulatory role of Hippo signaling on hemocyte phagocytosis was investigated using flow cytometry. Compared with the control, the phagocytosis of hemocytes in Hippo- and Wts-silenced shrimp was compromised but enhanced in Yki-silenced shrimp (Figure 4A–G). These suggest that Yki can suppress shrimp cellular immunity.

### 2.5. The Implication of Hippo–Yki Pathway in Antibacterial Response

Using RNAi strategy, we investigated the role of shrimp Hippo pathway components in *Vibrio* infection. The survival rates of *V. parahaemolyticus*-infected shrimp were significantly decreased after the silencing of Hippo and Wts but increased after the silencing of Yki (Figure 5A–C). Correspondingly, the content of *V. parahaemolyticus* in gills rose after Hippo and Wts silencing but declined after Yki silencing (Figure 5D). This indicates that the Hippo-Wts signaling positively regulates the antibacterial defense in shrimp, while Yki plays the opposite role.

## 3. Discussion

In this study, the expression profiles of Hippo pathway components in the immune response of *P. vannamei* have been investigated. After stimulations by fungi, Gram-positive and -negative bacteria, and the pathogen-associated molecular pattern (PAMP) of Gram-negative bacteria LPS, the upstream kinases of the Hippo pathway, Hippo and Wts, did not show significant expression changes. In contrast, Yki expression changed slightly after stimulation by most immune stimuli used in this study but increased significantly after *Vibrio* infection. This suggested that the Hippo–Yki signaling was involved in the immune defense against *Vibrio*. The limited change in Hippo and Wts expression and the significant increase in Yki expression may indicate the tendency of Hippo signaling inhibition and Yki activation in *Vibrio-*infected shrimp. At the transcription level, the shrimp Hippo–Yki pathway was associated with the Dorsal, Relish, and STAT pathways. The transcription of Hippo could be activated by STAT but inhibited by Dorsal, while that of Wts could be activated by both STAT and Relish. In contrast, the transcription of Yki could also be positively controlled by STAT and Relish but inhibited by Dorsal. The changes in their mRNA levels upon immune stimulations could be related to the regulatory effects of these immune signaling pathways. Whether these Hippo pathway components are also regulated by other signaling cascades requires further exploration. In turn, Hippo, Wts, and Yki could also regulate the expression of Relish, Dorsal, and STAT, indicating a complex regulatory network between Hippo–Yki signaling and the NF-κB and JAK-STAT pathways in shrimp that involves multiple positive and negative feedback loops and may play a role in the maintenance of immune homeostasis. Hippo is located upstream of Yki and is correlated with more factors [27]. The effect of Hippo silencing on Dorsal and STAT expression was opposite to that of Yki silencing, which conforms to the conduction rule of Hippo signaling. However, the regulation of Relish expression by Hippo and Yki showed the same trend. The mRNA level of Relish was down-regulated upon the silencing of Hippo and Yki, which may be caused by other unknown factors and need to be further explored.

At the protein level, the inhibition of Hippo-Wts signaling, that is, the activation of Yki, is marked by the dephosphorylation and nuclear translocation of Yki. After *V. parahaemolyticus* and *B. thuringiensis* infections, the levels of phosphorylated Yki protein were significantly reduced. Stimulation by *V. parahaemolyticus* showed a more significant effect on Yki dephosphorylation than that by *B. thuringiensis*. The duration of the Yki nuclear translocation responding to *B. thuringiensis* stimulation was relatively short, while its response to *V. parahaemolyticus* stimulation lasted more than 6 h. This suggested that Hippo signaling was inhibited upon bacterial infection in shrimp and showed different responses to Gram-negative bacteria and Gram-positive bacteria.

Down-regulation of the expression of Hippo and Wts in vivo by RNAi decreased the survival rate of *Vibrio*-infected shrimp and increased the bacterial content in tissues, while down-regulation of Yki expression showed the opposite result, indicating that the activation of Yki and the inhibition of Hippo signaling could promote *Vibrio* infection in shrimp. Studies in *Drosophila* have also revealed the response of Hippo signaling to bacterial infection; in the fat bodies of adult *Drosophila*, Hippo signaling is involved in the regulation of antimicrobial response [28,29]. Impairment of Hippo and Wts, i.e., overactivation of Yki, increased the susceptibility of *Drosophila* to Gram-positive bacterial and fungal infections by down-regulating the production of AMPs [28,29]. This indicates that the role of Hippo–Yki signaling in antibacterial responses could be different in shrimp and in *Drosophila*, the mechanism of which is worth in-depth exploration.

In mammals, the Hippo-signaling-mediated regulation of immune response is more complex because of the cross-talk between various immune-related signaling pathways [18,19,20]. Hippo, Wts, and Yki are implicated in innate immunity, inflammatory response, and, more importantly, antitumor immunity by regulating the production of cytokines and chemokines and the development, proliferation, and apoptosis of immune cells [12,13,14]. In this study, the inhibition of the Dorsal pathway by Yki may also establish the negative role of Yki in antibacterial immunity in shrimp. Moreover, invertebrates have only an innate immune system, which also consists of humoral immunity and cellular immunity. Hemocyte phagocytosis is an important part of cellular immunity in shrimp and plays a crucial role in pathogen elimination. We observed that Hippo signaling has a positive effect on hemocyte phagocytosis. The inhibition of Hippo and Wts weakened the phagocytic activity of shrimp hemocytes, while the inhibition of Yki enhanced it. The effects of these Hippo–Yki pathway components on hemocyte phagocytosis could be important factors for their roles in the anti-*Vibrio* response.

Although Yki negatively regulates the antibacterial response in shrimp, *V. parahaemolyticus* infection leads to the inhibition of Hippo signaling and the overactivation of Yki. This indicates that the regulation of Hippo–Yki signaling could be a strategy for *V. parahaemolyticus* to evade the immune defense of shrimp; this is worth further exploration. The mechanism by which *V. parahaemolyticus* regulates the intracellular Hippo–Yki signaling pathway is interesting and worthy of further exploration. This study shows the regulatory effect of *V. parahaemolyticus* on the activation of Hippo–Yki signaling, suggesting that there are specific cellular receptors in shrimp that mediate the recognition of *Vibrio* or its PAMP molecule LPS and conduct signals to the cell to regulate the activation of Hippo, Wts, and Yki. Studies in shrimp have revealed that some pattern recognition receptors (PRRs), such as leulectin and hemocyanin, are responsible for the recognition of bacteria and LPS [30,31]. Further investigation is needed as to which PRRs bridge *Vibrio* and the Hippo–Yki pathways. Furthermore, recent studies also suggest that the *V. parahaemolyticus*-secreted toxin PirAB with lectin activity can specifically recognize the beta-hexosaminidases subunit beta and mucin-like glycoprotein on hepatopancreatic epithelia and, putatively, the aminopeptidase N1 receptor (LvAPN1) on hemocytes [32,33]. Whether the *Vibrio* PirAB plays a role in the regulation of Hippo–Yki signaling through these receptors is also worthy of further exploration.

Taken together, the current work suggests that Hippo signaling is inhibited by bacterial infection in shrimp, leading to the overactivation of Yki. On the other hand, the activation of Hippo signaling and the suppression of Yki can significantly reduce the susceptibility of shrimp to *Vibrio* infection. Therefore, targeting the Hippo–Yki pathway can be utilized as a strategy for preventing *Vibrio* diseases in farmed shrimp.

## 4. Materials and Methods

### 4.1. Animal and Pathogens

Shrimp (~10 g) were purchased from a shrimp farm in Guangzhou, China, and cultured in recirculated and air-pumped seawater with a salinity of ~1% at ~27 °C. The *V. parahaemolyticus* were prepared as previously described [34].

### 4.2. qPCR

After challenges with *V. parahaemolyticus* (10^5^ CFU), Gram-positive *S. aureus* (10^5^ CFU), the fungus *A. niger* (10^5^ CFU), and LPS (5 μg), gills were extracted from nine shrimp at various time points and subjected to RNA purification to prepare cDNA. The specific primers to Hippo, Wts, and Yki were designed based on their sequence in Genbank (Accession No. ON778001, ON778002, and ON778003) and are listed in Appendix A. Real-time qPCR was performed with the EF1-α (Genbank accession No. GU136229) as the internal control using TB Green Premix Ex Taq II (Takara, Kyoto, Japan), following the manufacturer’s instruction. Reactions were done on the LightCycle 480 System (Roche, Mannheim, Germany), following previously described parameters [35].

### 4.3. Western Blot

Hemocytes from six shrimp were treated with SDS buffer and analyzed by Western blot using a rabbit anti-phosphor-Yki (S127) antibody (Cell Signaling Technology, Boston, MA, USA) for the phosphorylated Yki protein. The β-actin was detected with a specific antibody (MBL, Tokyo, Japan) as the internal control. Furthermore, the cytoplasmic and nuclear proteins of hemocytes were extracted using an NE-PER nuclear extraction kit (Thermo Fisher Scientific, Waltham, MA, USA) and analyzed using the anti-Yki antibody (GL Biochem, Shanghai, China), the anti-β-actin antibody, and the anti-histone H3 antibody (Cell Signaling Technology, Boston, MA, USA) to detect the internal nuclear control. The protein levels were determined based on the gray values of protein bands on the blot examined with Quantity One 4.6.2 software (Bio-Rad, Hercules, CA, USA) using the Gauss model and normalized to those of the control protein bands [36].

### 4.4. Immunofluorescence

After fixing with 4% paraformaldehyde for 10 min, shrimp hemolymph smears on siliconized slides were treated with 1% Triton X-100, blocked with 10% normal goat serum, incubated with rabbit antibodies against shrimp Yki (GL Biochem, China), and detected using Alexa Fluor 488-conjugated goat anti-rabbit IgG antibody (Abcam, Cambridge, UK). After staining with Hoechst 33342 (Sigma, St. Louis, MO, USA), smears were observed using a Leica LSM 410 confocal microscope (Germany).

### 4.5. Dual-Luciferase Reporter Assays

The promoter sequences of Hippo, Wts, and Yki were obtained from the *P. vannamei* genome in Genbank [37] and linked into the pGL3 plasmid with a firefly luciferase coding sequence (Promega, Madison, WI, USA). The shrimp Dorsal, Relish, and STAT expression plasmids were used as previously reported [38,39]. A mixture of 0.05 μg PGL3, 0.1 μg protein expression plasmid, and 0.03 μg pRL-TK Renilla luciferase plasmid (internal control, Promega, USA) was transfected into *Drosophila* S2 cells cultured in 96-well plates. At 48 h post transfection, cells were detected for the activities of firefly and Renilla luciferases with a dual-luciferase reporter assay kit (Promega, USA). Experiments were repeated with six independent transfections, and two-tailed unpaired Student’s *t*-tests were used to analyze the data.

### 4.6. Shrimp Challenge

The specific dsRNAs of Hippo (dsRNA-Hippo), Wts (dsRNA-Wts), Yki (dsRNA-Yki), Relish (dsRNA-Relish), Dorsal (dsRNA-Dorsal), STAT (dsRNA-STAT), and GFP (dsRNA-GFP, as control) were prepared using the T7 RiboMAX Express RNAi kit (Promega, Madison, WI, USA) and injected into shrimp, which were further challenged with *V. parahaemolyticus* (10^5^ CFU) at 48 h post injection. The survival rates were recorded and analyzed by the log-rank (Mantel–Cox) test. The content of *V. parahaemolyticus* in gills at 48 h post infection (hpi) was detected by qPCR, as previously described [36].

### 4.7. Phagocytic Activity Analysis

At 48 h post dsRNA injection, hemocytes were sampled, washed with PBS, pre-stained with Dil (Beyotime, Beijing, China), and incubated with fluorescein isothiocyanate (FITC)-labeled *V. parahaemolyticus* at a ratio of 1:100 for 1 h at 28 °C. Hemocytes were then analyzed by flow cytometry with 50,000 events detected for each sample. The fluorescence gate of cells that have phagocytized bacteria was set based on the fluorescence of the FITC-*Vibrio*, unstained and Dil-stained hemocytes.

### 4.8. Statistical Analysis

Different batches of shrimp were used to perform the experiments three times. Two-tailed unpaired Student’s *t*-tests or one-way ANOVAs followed by Dunnett’s post hoc test were performed to analyze the statistical comparisons. The results were representatively demonstrated, and the data are provided as mean ± standard deviation (SD). The survival rate data were analyzed by Kaplan–Meier log-rank χ^2^ tests.

## Figures and Tables

**Figure 1 ijms-23-11897-f001:**
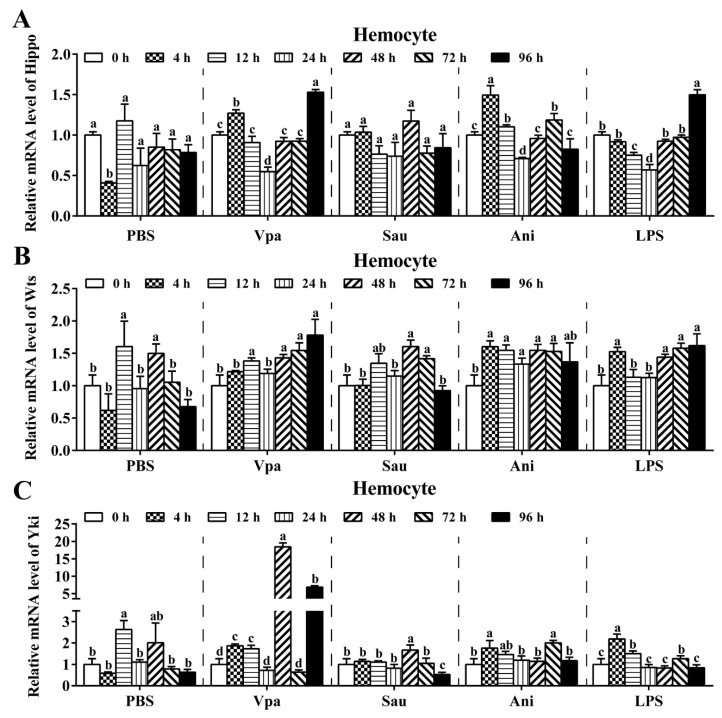
Expression of the Hippo signaling components in shrimp infected with bacteria and fungi. The mRNA levels of Hippo (**A**), Wts (**B**), and Yki (**C**) in hemocytes after *V. parahaemolyticus* (Vpa), *S. aureus* (Sau), *A. niger* (Ani), LPS, and the control PBS were detected using relative qPCR. For each challenge group, the data at 0 h was set as 1.0 after normalizing to the geometric mean expression of the internal controls β-actin and Ef-1α. Values with different letters indicated significant differences by one-way ANOVA, followed by Dunnett’s post hoc test (*p* < 0.05).

**Figure 2 ijms-23-11897-f002:**
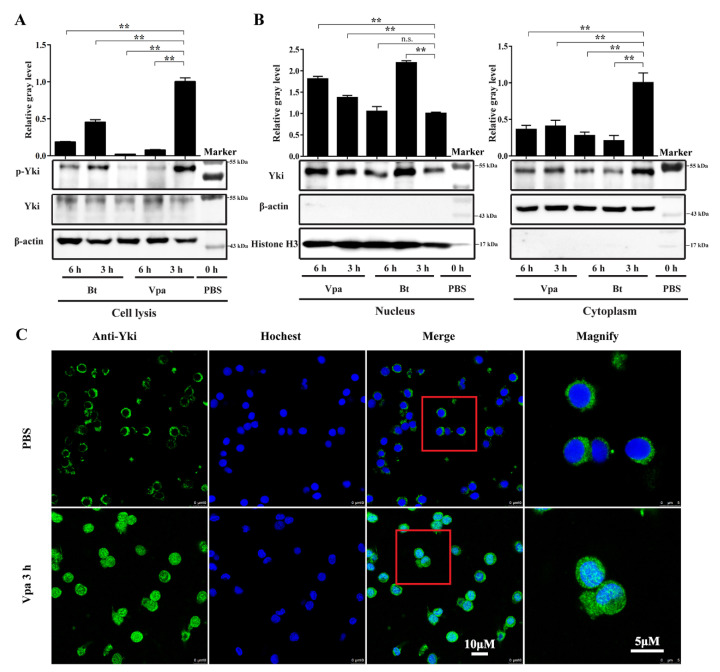
Activation of Yki after bacterial infection. (**A**) Western blot analysis of phosphorylated and unphosphorylated Yki in hemocytes from shrimp infected with *V. parahaemolyticus* (Vpa) and *B. thuringiensis* (Bt). The gray values of Yki bands were normalized to those of β-actin. (**B**) Western blot analysis of Yki protein distribution in the nucleus and cytoplasm of hemocytes from bacteria-infected shrimp. The gray values of Yki bands in the nucleus and cytoplasm were normalized to those of histone H3 and β-actin, respectively. **, *p* < 0.01 and n.s., *p* > 0.05 by two-tailed unpaired Student’s *t*-test. (**C**) Immunofluorescence analysis of the expression of Yki in hemocytes at 3 h post *V. parahaemolyticus* or PBS-mock infection.

**Figure 3 ijms-23-11897-f003:**
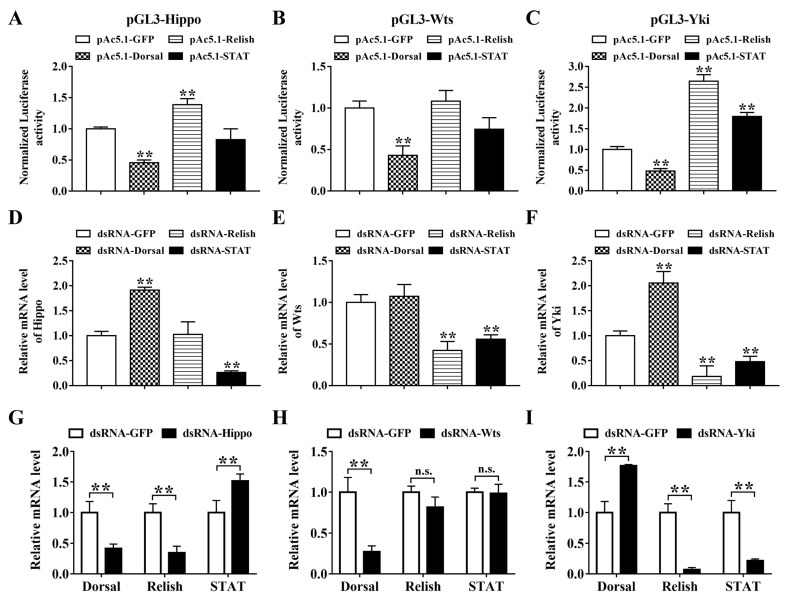
Regulatory relationship between Hippo signaling components and NF-κB/STAT. Regulatory effects of Relish, Dorsal, and STAT on the promoters of Hippo (**A**), Wts (**B**), and Yki (**C**) were examined by dual-luciferase reporter assay. The mRNA levels of Hippo (**D**), Wts (**E**), and Yki (**F**) in Dorsal-, Relish-, and STAT-silenced shrimp and those of Dorsal, Relish, and STAT in Hippo- (**G**), Wts- (**H**), and Yki-silenced (**I**) shrimp were investigated by qPCR. The gene knockdown efficiencies in vivo are shown in Appendix A. ** *p* < 0.01 and n.s. *p* > 0.05 by two-tailed unpaired Student’s *t*-test.

**Figure 4 ijms-23-11897-f004:**
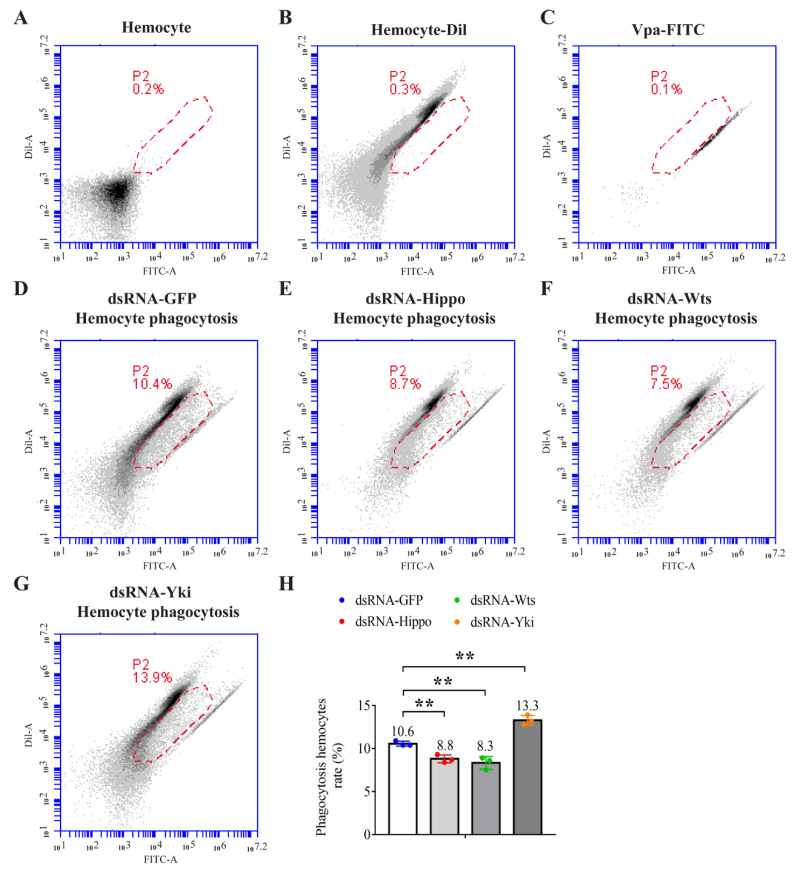
Regulation of hemocyte phagocytosis by the Hippo pathway. (**A**–**C**) Determination of the fluorescence signal gate of phagocytized bacteria in hemocytes by analyzing the unstained (**A**) and Dil-stained hemocytes (**B**) and FITC-labeled *V. parahaemolyticus* (Vpa, **C**) by flow cytometry. The phagocytic activity of hemocytes of dsRNA-GFP- (**D**), dsRNA-Hippo- (**E**), dsRNA-Wts- (**F**), and dsRNA-Yki-treated shrimp (**G**) were analyzed and statistically analyzed (**H**). **, *p <* 0.01 by two-tailed unpaired Student’s *t*-test.

**Figure 5 ijms-23-11897-f005:**
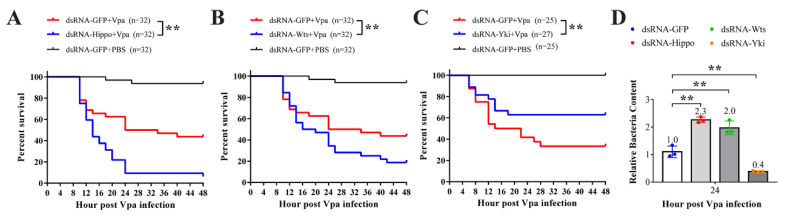
The implication of the Hippo pathway in *Vibro* infection. The survival rates of *V. parahaemolyticus-*infected shrimp after silencing Hippo (**A**), Wts (**B**), and Yki (**C**) were statistically analyzed. (**D**) The *V. parahaemolyticus* bacterial load in gills was detected by qPCR. ** *p* < 0.01 by two-tailed unpaired Student’s *t*-test.

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
