# Peer review of "The Hippo–Yki Signaling Pathway Positively Regulates Immune Response against Vibrio Infection in Shrimp"

_ijms, 2022, doi:10.3390/ijms231911897_

Round 1
Reviewer 1 Report
In this manuscript, the authors investigated the influence of the Hippo-YAP pathway in pacific white shrimp upon bacterial infection. They find that upon infection, Yap is dephosphorylated and thus activated and translocates to the nucleus. This suggests an inhibition of Hippo which acts to phosphorylate YAP and sequester it in the cytoplasm. The author’s further find that the activation of YAP is actually detrimental to the shrimp when infected, as when YAP is knocked down, the shrimp live longer after bacterial infection.
The manuscript overall did not seem to be very focused and did not come to clear conclusions, with contradictory figures. It is also difficult to interpret many figures as there is no information of the number of biological replicates and statistics used. Below I have highlighted some of the major concerns of the paper:
Figure 1 is uninterpretable. It is really unclear what is being compared for the statistics, and what the letters actually mean. Further, there is huge variability in the pbs treated Hemocytes making it unclear how to even interpret the data. The author’s point out the significant increase in YAP expression at 48 and 96 hours, but why then is 72 hours so low. Further, if it takes 48h or 96 hours to see a response why are the rest of the experiments done at 3 and 6 hours? It is also not clear why gene expression is even being looked at, as these are proteins that interact through post translational modifications so it is not clear that looking at mRNA levels is particularly useful. The author’s need to include the N value in the figure legend. Because qPCR can be variable it is best practice to normalize data to 3 housekeeping genes and not just 1.
Figure 2. Is panel B phosphorylated YAP or YAP? According to the legend, it is phosphorylated YAP, but then there should not be any phosphorylated YAP in the nucleus. It would be best to show both phosphorylated YAP and YAP in both A and B. Molecular markers should also be included in the westerns. What is the N value of biological replicates for these experiments and what was the statistical test used? When showing non-significant comparisons please write n.s.
It is not clear how figure 3 relates to the paper overall. In figure 1 the authors show that there is essentially no changes in Hippo, Wts of YAP (except at 48h and 96h after Vpa infection). If NF-kB/STAT pathway played a role in regulating the transcription of these genes in a natural infection, it would be anticipated that changes would have been seen in figure 1. Again, what is the N-value and what statistics were used for significance?
In the discussion, I would like the authors to speculate why, if YAP plays a negative role in fighting off bacteria it would be upregulated?
Reviewer 2 Report
The manucsript « The Hippo-YAP Signaling Pathway Positively Regulates Immune Response Against Vibrio Infection in Shrimp » by Yang Ref. ijms-1868670 is an interesting work that aims to identify the mechanisms that bacteria of the genus Vibrio have to regulate the response of crustaceans, particularly in shrimps. Some tests are carried out to identify the activation processes of transduction pathways and particularly the focus on Hippo and YAP draws attention. The results and the approach have some significant data, however, some aspects must be clarified..
1. An extensive evaluation of the effect on Hippo-YAP signaling by Vibrio is made and it is compared with various bacteria, it would be important for the authors to indicate what the objective of this comparison is, the participation of common activating receptors for some of these PAMPs is assumed ?. It is not very clear why compare bacteria vs LPS as endotoxin specific for TLR4?
2. The biological effects analyzed are carried out in samples of up to 96 hours and in fact YAP is affected in processes greater than 48 hours. Is it important to discuss this result, is it due to the persistence and not elimination of the pathogen? Are they secondary effects to the infectious stimulus? It would be interesting if the authors could demonstrate if there is (even in such long evaluation periods) morpho-functional alteration of the cell populations. At least blood count and viability would be a good option.
3. Crustaceans have a limited number of hyaline and granular cell lines, which group of cells are used to evaluate phagocytosis? Is a total population used?
4. What is the rationale for evaluating the Activation of the YAP after bacterial infection, using B. thuringiensis???fig 2
5. It is again interesting to identify that the YAP activation pathways in figure 2 are identified changes up to 6h, what is then the value that is given to the effect shown at 96h after infection?
6. There is various evidence suggesting that the interaction of V. parahaemolyticus is specific for glycosylated receptors in the tissue to be infected. It would be interesting to know some perspective proposed by the authors in this regard, which could also justify the use of various pathogens, including LPS.
7. Figure 4, the relationship with work in general is not clear, the results or had should be better supported in the discussion.
8. Finally what is the conclusion? Is HIPPO-YAP regulation evident in the specific response to a particular pathogen such as V. parahemolyticus but not to others? The possibility of specific interaction of Vibrium with receptors not PAMPv.gr other glycosylated receptors) is what generates the effect?.
Round 2
Reviewer 1 Report
The author's have made improvements and satisfactorily addressed my concerns
Author Response
Thank you very much.
Reviewer 2 Report
In relation to the manuscript re ijms-1868670, unfortunately in this version my main concerns have not yet been resolved. It is not working with specific cell populations, after the treatments there could be a change in the proportion of some of the cell populations (granular, hyaline or semigranular) that have specialized biological activities and that, in the context of the responses evaluated during so many hours There should be a parameter at least morphological that allows certainty in the differences found as an effect of the bacterial stimulus. Explanations for the use of different bacterial strains are not convincing, the structural characteristics of Gram positive or negative bacteria are important and this would need to be discussed or justified.
Author Response
Thank you for your suggestion. It has been known that shrimp hemocytes are divided into three main subpopulations: granular, hyaline and semi-granular cells, which can be observed by confocal microscopy (Junprung W, Supungul P, Tassanakajon A. Dev Comp Immunol. 2019, 90:138-146). In the revised manuscript, the immunofluorescence assay showed that the Yki protein could be detected in all three types of hemocytes (Figure 2C, Bright-field). The Yki protein was mainly present in the cytoplasm of hemocytes from unstimulated shrimp. However, at 3 h post V. parahaemolyticus infection, the expression of Yki was obviously increased and more Yki proteins were present in the nuclei of all hemocytes (Figure 2C), indicating the activation of Yki upon V. parahaemolyticus stimulation. These suggest that the Hippo-Yki pathway functions in all three types of hemocytes, in which it is involved in the regulation of immunity and homeostasis. This information and reference have been added in lines 100-107 and discussed in lines 245-251 of the revised manuscript.
For the Gram-positive and -negative bacteria issue, in the Discussion section of the revised manuscript, the following paragraph has been added in lines 212-223: ‘At the protein level, inhibition of Hippo-Wts signaling, that is, activation of Yki, is marked by dephosphorylation and nuclear translocation of Yki. After V. parahaemolyticus and B. thuringiensis infections, the levels of phosphorylated Yki protein were significantly reduced. The stimulation by V. parahaemolyticus showed a more significant effect on Yki dephosphorylation than that by B. thuringiensis. The duration of the Yki nuclear translocation responding to B. thuringiensis stimulation was relatively short, while its responding to V. parahaemolyticus stimulation lasted more than 6 hours. These suggested that the Hippo signaling was inhibited upon bacterial infection in shrimp and showed different responses to Gram-negative bacteria and -positive bacteria. The difference in the activation of Hippo pathway by these two types of bacteria may be due to their differences in cell wall structure and PAMP molecules, which is worthy of further study.’
Round 3
Reviewer 2 Report
Unfortunately, although the version of the manuscript is better improved, it does not confirm the regulatory role of the interaction with pathogenic bacteria for crustaceans such as V. parahemolyticus. It would be interesting if the authors in further studies considered that the bacterial interaction could be regulated by the participation of specific receptors rather than due to the interaction of the components of the bacterial glycocalyx, as suggested by the interaction with PAMPS from B. Thuringiensis.